# Predictors of Liver Injury in Hospitalized Patients with SARS-CoV-2 Infection

**DOI:** 10.3390/medicina58121714

**Published:** 2022-11-23

**Authors:** Nicoleta Mihai, Mihai Lazar, Catalin Tiliscan, Ecaterina Constanta Barbu, Cristina Emilia Chitu, Laurentiu Stratan, Oana Alexandra Ganea, Sorin Stefan Arama, Daniela Adriana Ion, Victoria Arama

**Affiliations:** 1Faculty of Medicine, “Carol Davila” University of Medicine and Pharmacy, 37 Dionisie Lupu Street, 020021 Bucharest, Romania; 2“Prof. Dr. Matei Bals” National Institute for Infectious Diseases, 1 Dr. Grozovici Street, 021105 Bucharest, Romania

**Keywords:** liver injury, SARS-CoV-2, risk factors, COVID-19 complications

## Abstract

*Background and Objectives*: SARS-CoV-2 infection is frequently associated with pneumonia but has a broad tissue tropism also leading to systemic complications (hematologic, gastro-intestinal, cardiac, neurologic, etc.). In this study, we aim to evaluate the impact of COVID-19 infection on the liver and to identify the risk factors/predictors for liver injury at admission to the hospital. *Materials and Methods*: We performed a retrospective cohort study on 249 patients, divided into two Group A (157 patients with liver involvement) and Group B (92 patients without liver involvement). We recorded demographic and lifestyle parameters, anthropometric parameters, comorbidities, clinical parameters, inflammation markers, complete blood count, coagulation, and biochemical parameters. Lung parenchyma, liver dimensions, and morphology were evaluated by computer tomography (CT) scans. *Results*: Patients with liver involvement had higher heart and respiratory rates, lower oxygen saturation (SO_2_), and necessitated higher oxygen flow at admittance. We found higher serum levels of C-reactive protein, fibrinogen, ferritin, creatine kinase, lactate dehydrogenase (LDH), serum triglycerides, and lower values for serum albumin in Group A patients. The patients with liver involvement presented more extensive lung injury with higher percentages of alveolar, mixed, and interstitial lesions, an increase in liver dimensions, and lower density ranges for the liver parenchyma. The patients presented hepatocytolytic involvement in 26 cases (10.4% from the entire study population), cholestatic involvement in 63 cases (37.7% from the entire study population), and mixed liver involvement in 68 cases (37.7% from the entire study population). *Conclusions*: Liver involvement in COVID-19 patients is frequent, usually mild, and occurs mostly in male patients over 50 years old. Cholestatic and mixed liver injuries are more frequent than hepatocytolytic injuries. The severity of lung injury evaluated by CT scan, increased values of inflammatory markers, LDH, and low values of SO_2_ can be considered risk factors/predictors for liver injury at admission to the hospital.

## 1. Introduction

COVID-19 represents an important worldwide health problem, with over 610 million cases and 6.5 million deaths registered in the last 30 months [1]. SARS-CoV-2 mainly induces respiratory symptomatology frequently associated with pneumonia but has a broad tissue tropism also leading to systemic complications (hematologic, gastro-intestinal, cardiac, neurologic, etc.) [2]. Liver involvement has also been reported in COVID-19 patients [3], with prevalence ranges between 3.75% [4] and76.3% [5].

There are several hypotheses regarding the mechanism of liver injury in COVID-19 patients. The main culprits seem to be the systemic inflammatory response syndrome and drug toxicity [6]. Several authors have shown an association between liver injury and elevated inflammatory markers, such as C-reactive protein (CRP) and ferritin [7,8]. Furthermore, a correlation between liver injury and some of the drugs included in the COVID-19 therapeutic guidelines has already been demonstrated in previous studies [5,9]. Pre-existing liver disease and hypoxia can also be involved, as well as direct viral damage [3]. Furthermore, according to previous studies, neutrophilia, lymphopenia, elevated CRP and male gender may predict liver damage in COVID-19 patients [10,11].

The studies carried out so far have shown that liver injury can be associated with severe COVID-19 [12]. Moreover, increased values of liver enzymes, especially the AST level, are associated with a higher risk of mortality [10].

However, the mechanism of liver injury and the predictors for liver involvement in COVID-19 patients are not yet fully understood and studies to confirm or refute the existing hypotheses are still needed.

In this research, we aimed to evaluate the impact of COVID-19 infection on the liver and to identify the risk factors/predictors for liver injury at admission in hospital.

## 2. Materials and Methods

In our retrospective cohort study, we enrolled 249 patients diagnosed with COVID-19, admitted in our department between January 2021 and May 2021, and we divided them in two groups: Group A (157 patients with abnormal liver function/injury) and Group B (92 patients without abnormal liver function/injury).

The study was approved by the local Ethics Committee of “Prof. Dr. Matei Bals” National Institute for Infectious Diseases (C14730/2021).

All patients were diagnosed with COVID-19 following a positive real time-polymerase chain reaction test (RT-PCR).

Abnormal liver function was considered present in patients with:increased serum level of alanine transaminase (ALT > 50 IU/L) or aspartate transaminase (AST > 59 IU/L) without exceeding three times the upper limit of normal (ULN);increased serum values of gamma-glutamyl transferase (GGT > 49 IU/L), alkaline phosphatase (ALP > 130 IU/L), or total bilirubin (TB > 1.2 mg/dL) without exceeding twotimes ULN.

We considered liver injury to be present in patients with ALT or AST exceeding three times ULN and ALP, GGT or TB exceeding 2 times ULN [6,12].

Considering the information mentioned above, we defined 4 grades for the liver involvement in patients with COVID-19, depending on the ULN for the liver enzymes:-grade 1 (1–3 ×ULN for ALT/AST; 1–2 × ULN for ALP, GGT or TB);-grade 2 (3–5 × ULN for ALT/AST; 2–4 × ULN for ALP, GGT or TB);-grade 3 (5–10 × ULN for ALT/AST; 4–8 × ULN for ALP, GGT or TB);-grade 4 (≥10 × ULN for ALT/AST; ≥8 × ULN for ALP, GGT or TB) [9].

Grade 1 corresponds to abnormal liver function, grades 2 to 4 correspond to liver injury.

We considered the liver involvement as hepatocytolytic for the patients with elevated ALT and/or AST only, cholestatic for the patients with increasead serum levels of ALP, GGT and/or TB, and mixed in the patients with increases of both types of liver enzymes [6].

All patients in Group B presented normal ALT, AST, ALP, GGT and TB serum values.

The inclusion criteria in the study were: age over 18 years, positive diagnosis of COVID-19 by RT-PCR. We applied following exclusion criteria: presence of pregnancy, acute or chronic hepatitis of any other cause (A, B, C and D viral hepatitis, Epstein–Barr virus infection, cytomegalovirus viral infection, alcoholic hepatitis, autoimmune hepatitis), documented hepatic steatosis, primary or secondary hepatic malignacy, partial hepatectomy, treatment with cytostatic or any known hepatotoxic medication.

### 2.1. Registered Parameters in the Study Participants

We recorded demographic and lifestyle parameters (age, sex, smoking status, alcohol intake status), anthropometric parameters (weight, height, body mass index (BMI)), and comorbidities (diabetes mellitus, chronic pulmonary disease, chronic kidney disease, malignacy, HIV infection, symptomatology (fever, chills, cough, nasal congestion, rhynorrhea, dyspnea, polypnea, thoracic and abdominal pain, nausea, vomiting, loss of appetite, dysphagia, anosmia, ageusia, diarrhea, asthenia, myalgia, arthralgia, conjunctival hyperemia, headache, paresthesia, confusion, convulsion, coma).

We documented clinical parameters (heart rate, systolic and diastolic blood pressure, respiratory rate, SpO_2_ by pulse oximetry, oxygen flow, days of hospitalization), inflammation markers [C-reactive protein (CRP), serum ferritin, erythrocyte sedimentation rate (ESR), interleukin 1 (IL-1), interleukin 6 (IL-6), tumor necrosis factor alpha (TNFα), white blood cells (WBC)], complete blood count, and coagulation parameters [D-dimers, plasminogen activator inhibitor-1(PAI-1), fibrinogen, prothrombin time (PT)].

Following biological serum parameters were measured: N-terminal pro B-type natriuretic peptide(NT proBNP), troponin I (TnI), creatine kinase including MB isoform (CK, CKMB), myoglobin, serum proteins, serum albumin, lactate dehydrogenase (LDH), alanine transaminase, aspartate transaminase (ALT, AST), GGT, ALP, direct, indirect and total bilirubin (DB/IB/TB), serum urea and serum creatinine, serum lipase, glycaemia, glycated hemoglobin (HbA1c), serum triglycerides (TG), total and fractionated serum lipoproteins.

### 2.2. Radio-Imaging Evaluation in the Study Participants

All patients were evaluated by computer tomography (CT) scans with a 64-slice Definition AS (Siemens Healthcare GmbH) within 48 h of admission. The patients were examined in inspiratory breath-hold and supine position. A dedicated diagnostic software, syngo Pulmo3D was used for the quantitative evaluation of lung parenchyma. The following factors were considered: alveolar lesion–the lung areas with densities higher than 0 Hounsfield units (HU), mixed lesions (alveolar and interstitial)–the lung areas with densities between 0 and −200 HU, interstitial lesions–the lung areas withdensities between −200 and −800 HU, and normal parenchyma–the lung areas with densities between −800 and −1000 HU [13]. Additionally, we measured the cranio-caudal diameter of the right hepatic lobe (RHL), antero-posterior diameter of the left hepatic lobe (LHL) and liver density (LD) (three measurements were performed in both hepatic lobes and the mean value was recorded; the measurements were consistent for all patients, maintaining the same dimension of the measured density area).

The imaging review was blinded; the radiologist was unaware of any parameters registered for the patient or study group enrolment.

### 2.3. Statistical Analysis

For the statistical analysis, we used Statistical Package for Social Sciences (SPSS version 25, IBM Corp., Armonk, NY, USA). Patient data are presented as medians and quartiles (Q1, Q3) for continuous variables and as percentages for the categorical variables. For the continuous variables we used the Mann–Whitney test and for categorical variables we used the Chi-square test. We performed binary logistic regression using liver involvement as dependent variable and the demographic, clinical, biologic, and imaging data as independent variables, presenting the odds ratio (OR) and the 95% confidence interval (95%CI) for every recorded variable. We used OR to estimate the risk of liver involvement for the independent variables: for OR > 1 the correlation between liver involvement and independent variable is proportional, and the liver involvement risk is calculated using the formula (OR − 1) ×100%. To calculate “double risk” we used the formula 1/(OR − 1). We calculated the relationship between liver involvement and the registered patient data using Pearson correlation. A *p*-value lower than 0.05 was considered statistically significant.

## 3. Results

### 3.1. Characteristics of the Study Participants

We enrolled in Group A 157 patients—110 males with a median age of 50 [42; 59.2] years and 47 females with a median age of 61 [52; 71] years, with a sex ratio male:female of 2.34:1.

Group B included 92 patients—47 males with a median age of 52 [43; 69] years and 45 females with a median age of 62 [49.5; 71] years, with a sex ratio male:female of 1.04:1.

The patients in Group A were slightly younger than the patients in Group B, 53 [43.5; 63.5] years vs. 56.5 [45.2; 70] years, registering a higher percentage of males, 70.1% vs. 51.1%.

The percentages of comorbidities were slightly higher in Group B with exception of overweight and obesity which were increased for patients in Group A, but these differences were not statistically significant (Table 1).

### 3.2. Clinical, Laboratory and Imaging Data of the Patients with and without Liver Involvement

Patients with liver involvement had higher heart and respiratory rates, lower oxygen saturation and necessitated higher oxygen flow at admittance. We found higher serum levels of CRP, fibrinogen and ferritin in Group A patients, suggesting a more intense inflammatory process. Aside fromincreased liver enzymes (ALT, AST, GGT, ALP, BT, BD), the patients in Group A presented also higher serum values for CK, LDH, TG and lower values for serum albumin. We found in COVID-19 patients with liver involvement a more extensive form of lung injury with higher percentages of alveolar, mixed and interstitial lesions, an increase in liver dimensions and lower density ranges for the liver parenchyma (consistent with increased liver steatosis) (Table 2).

For an increase of1unit in the parameters (Table 2) registered for the patients in Group A, the risk of developing liver involvement will increase with 2% for the heart rate, with 17% for the respiratory rate, with 17% for the oxygen flow at admission, with 0.7% for CRP, with 0.2% for serum ferritin, with 0.3% for fibrinogen, with 0.2% for CK, with 0.7% for LDH, with 12% for ALT, with 15% for AST, with 16% for GGT, with 3% for ALP, with 2530% for DB, with 320% for DB, with 1% for TG, with 88% for the percentage of lung consolidation, with 51% for the percentage of mixed lung lesions, with 8% for the percentage of interstitial lesions, with 7% for the percentage of total pulmonary lesions, with 69% for the number of pulmonary lobes with pneumonia, with 14% for the number of pulmonary segments with pneumonia, with 29% for RHL. For a decrease with 1 unit in the parameters registered for the patients in Group A, the risk of developing liver involvement will increase with 14% for saturation, with 67% for serum albumin, with 6% for normal pulmonary densities, with 4% for the liver density.

Considering the results presented above, the patients may face a double risk of developing liver involvement for every increase (of the median values for Group A in Table 2) in heart rate with 50 beats/min, respiratory rate with 6 breaths/minute, oxygen flow at admission with 6L/min, CRP with 143 mg/L, serum ferritin with 500 ng/mL, fibrinogen with 333 mg/dL, LDH with 143 mg/dL, and CK with 500 IU/L, ALT with 8 IU/L, AST and GGT with 7 IU/L, ALP with 33 IU/L, DB with 0.04 mg/dL, TB with 0.24 mg/dL, and TG with 100 mg/dL. Every decrease (of the median values for Group A in Table 2) in SO_2_ with 7 and in serum albumin with 1.5 g/dL is also associated with a double risk of developing liver involvement.

Analyzing the results of the imaging evaluation, the patients with COVID-19 present a double risk of liver involvement for every increase (of the median values for Group A in Table 2) in percent of lung consolidation with 1.13%, percent of mixed lung lesions with 1.96%, percent of interstitial lung lesions with 12.5%, percent of total pulmonary lesions with 14.3%, dimension of RHL with 3.4 cm. Every decrease (of the median values for Group A in Table 2) in percentage of normal pulmonary densities with 16.6% and in liver density with 24HU is also doubling the risk to develop liver involvement.

### 3.3. Correlations of Clinical, Laboratory and Imaging Parameters with Liver Involvement

The liver involvement in patients with COVID-19 presented the highest proportional correlation with the radio-imaging parameters (percent of total pulmonary lesions, percent of interstitial pulmonary lesions) and LDH (Table 3). Liver involvement was also correlated with increased serum values in inflammatory markers (CRP, fibrinogen, serum ferritin), with higher respiratory and heart rate, higher TG levels and lower serum albumin. The severity of lung injury evaluated by CT scan presented a correlation with liver involvement superior to liver morphological data (RHL, LHL and LD). In addition to data presented in Table 3, we found proportional correlations with statistical significance between the liver involvement and weight of the patient (0.183) and male gender (0.19).

The symptoms found in patients at admission are presented in Table 4. Comparing both study groups, the patients in Group A presented more frequently fever (80.3% vs. 63%), diarrhea (28% vs. 15.2%), headache (45.2% vs. 30.4%), polypnea (21% vs. 7.6%) and respiratory insufficiency (67.5% vs. 42,4%), while in Group B were more frequent the cough (89.1% vs. 81.5%), chills (51.1% vs. 44.6%), asthenia (64.1% vs. 57.3%) and myalgia (56.5% vs. 45.2%).

The overall mortality in the study was 0.8% (2 patients), without significant difference between the two groups: the mortality registered in Group A was 0.6% (1 patient) and in Group B 1.1% (1 patient).

### 3.4. Characteristics of Liver Involvement in the Study Participants

Liver involvement at admission was mostly mild, with abnormal liver function in 94 patients (representing 59.8% from Group A and 37.7% from entire study population) and liver injury in 63 patients (representing 40.2% from Group A and 25.3% from entire study population) (Table 5). The patients presented hepatocytolytic involvement in 26 cases (representing 16.5% from Group A and 10.4% from entire study population), cholestatic involvement in 63 cases (representing 40.1% from Group A and 37.7% from entire study population), and mixed liver involvement in 68 cases (representing 43.3% from Group A and 37.7% from entire study population).

Although the overall number of patients with increased ALT was higher than the number of patients with increased AST, when analyzing AST/ALT ratio we found values >1 in 88 patients (representing 56.1% from Group A and 35.3% from the entire study population) and values <1 in 68 cases (representing 43.3% from Group A and 27.3% from entire study population).

Comparing the characteristics of grade 1 (patients with abnormal liver function) to grade 4 (patients with liver injury), (Table 6) it can be observed that patients with liver injury grade 4 were obese or overweight males, with a longer hospitalization period, a more evident inflammatory syndrome (CRP, serum ferritin, ESR, IL-6), presenting metabolic disorders (TG, glycemia, HbA1C with higher serum values), more important coagulation/fibrinolysis changes (higher values for PAI-1 and fibrinogen) and higher LDH values than the patients with grade 1 with abnormal liver function.

Patients in both grades (1 and 4) presented similar lung injury percentage and similar liver dimensions and densities.

Analyzing the liver enzymes used in the grading process, it can be observed a more important increase (compared to ULN) in GGT, ALP and DB than in AST and ALT.

## 4. Discussion

In our study, 62.2% of patients included in the study population were older than 50 years (59.9% of patients in Group A and 66.3% in Group B), consistent with previous communicated data in SARS-CoV-2 infected patients [14,15]; although the patients in Group A were slightly younger than the patients in Group B, we didnot find any correlation between age and liver involvement.

We also found a higher proportion of males (sex ratio male:female = 1.7:1). Although, the mechanism is not explained, some authors consider that male sex hormones may increase the expression of ACE2, with a higher risk of coronavirus infection and a more severe form of disease [16]. Moreover, in our study, male gender was correlated with the occurrence of liver damage. This is in accordance with the currently published data, which showed that male gender is a risk factor for liver involvement in COVID-19 patients [10,17].

Regarding the liver assessment, 63% of patients presented liver involvement, considered abnormal liver function in 37.7% cases and liver injury in 25.3% cases. Although ACE2 has a high expression in bile duct cells and a high serum value of ALP and TB could be expected, in our study the serum values for ALT and AST were superior to biliary enzymes, consistent with information communicated by other authors [18,19,20].

Although ALT and AST are considerate liver enzymes, they also may be found in extrahepatic tissues [21,22,23], therefore patients with severe forms of COVID-19 pneumonia may be expected to have increase in serum transaminase values; however, the presence of pneumonia alone cannot explain the increased transaminase values in patients with mild or moderate forms of COVID-19, associated in some cases with increase in ALP, TB or GGT, consistent with data published by Huang et al. [24].

Overweight and obesity represent risk factors for COVID-19 hospitalization and prolonged evolution [25] which can explain the high percentage of patients in the study population with obesity (36.5%) and overweight (31.7%) and also the proportional correlation of liver involvement with increased TG serum values. In patients with increased BMI, the body mass composition may be associated also with increased transaminase values in case of severe liver steatosis and steatohepatitis [26,27,28].

Multiple comorbidities may induce an increase of liver enzymes, therefore, to explore more accurately the impact of SARS-CoV-2 infection on liver, the most frequentcauses were excluded (acute or chronic hepatitis, documented hepatic steatosis, autoimmune disease, primary or secondary hepatic malignacies, treatment with cytostatic or any hepatotoxic medication).

Liver injury is considered multifactorial and includes an active replication of coronavirus in the hepatocytes, systemic inflammation with multiorgan involvement, immune dysfunction, hypoxia-reperfusion injury and drug-induced liver injury (DILI) [29,30,31]. However, by evaluating the patients at admission, prior to COVID-19 treatment, and excluding from study the patients under hepatotoxic medication, a drug induced liver injury pathway may be excluded for the patients in our study. In contrast, patients with liver involvement had higher respiratory rates, lower oxygen saturation and required higher concentration oxygen therapy on admission, which underlines the role of hypoxia in the occurrence of liver injury.

The coagulopathy associated with COVID-19, usually diagnosed by high D-dimers levels, may also involve hepatic circulation. Sonzoni et al. [32] reported portal or sinusoidal thrombosis in at least 50% of patient with COVID-19. Activation of coagulation may also promote an increase in liver steatosis [33], frequently found in patients with COVID-19, consistent with the findings in our study. Experimental studies performed on mice demonstrated that activation of coagulation may also promote an increase in liver steatosis [33], however it is yet unclear if this is also the case in humans.

The presence of hepatic steatosis, frequently present in patients with COVID-19, also found in our study, could reveal a previously undiagnosed comorbidity of these patients.However, in a study conducted by Chen et al. on 830 patients with COVID-19, the liver steatosis found on the initial CT gradually recovered on follow-up CT scans during hospitalization, as the viral infection improved, the authors assuming that hepatic steatosis is a transient change associated with COVID-19 [34]. Microvesicular and macrovesicular steatosis was also described in post-mortem histological examination of liver biopsy samples [35].

There are also reports that suggest a possible association of endotheliopathy in the portal vein branches and hepatic artery branches in the portal tract with cholestasis [36,37]. The cholestatic and mixed liver injury type represented over 75% in the entire study population, an important aspect that necessitate long term monitorization due to the fact that cholestasis has been reported also as a long term complication of COVID-19, with possible need of liver transplant [37,38]. A persistent inflammatory state and immune activation, characterized by increased blood levels of the inflammatory cytokines–TNFα, IL-6, IL-1 and CRP, although not specific to COVID-19 [39], may induce apoptosis with secondary endothelial dysfunction and may increase intrahepatic clot development risk. In our study, we found that inflammatory markers (C-reactive protein, fibrinogen and ferritin) correlate with liver damage, which is also consistent with the existing data [11]. To improve the inflammatory storm, and thus a possible risk of liver involvement, corticosteroid therapy was administered in patients with severe cases of COVID-19 which showed improvements in the overall patient status and a reduced length of disease [40]. However, in patients with long hospitalization periods and prolonged steroid administration, complications of steroid administration (gastritis, pancreatitis, systemic arterial hypertension, hyperlipidemia, hepatic steatosis, peptic ulcer, bone loss, etc.) [41,42] should be considered, which may accentuate preexistent conditions and pathologies.

Several studies showed that lymphopeniaand neutrophilia are predictors for liver damage in COVID-19 patients [10,17]. In contrast, we did not find any correlation between lymphocytes, neutrophils or neutrophils/lymphocytes ratio (Neu/Ly) and liver involvement.

Liver involvement at admission was mostly mild, with liver injury in 63 patients (25.3% from entire study population), in 8% of cases the patients presenting grade 3 and 4 injuries. An interesting aspect in our study revealed by the Pearson analysis was a higher correlation with the hepatic involvement for the lung injury evaluated by CT scan (a coefficient of 0.335 for percent of interstitial lung lesions, 0.337 for the percent of total pulmonary lesions, 0.311 for the number of pulmonary lobes with pneumonia, and −0.337 for the percent of normal pulmonary densities) than for the liver measurement (0,23 for RHL, 0.144 for LHL and −0.269 for LD), consistent with findings of other authors [43], which communicated increased incidence of liver injury in severe cases of COVID-19 pneumonia.

*Study limitations*: The enzymes used to diagnose liver involvement in COVID-19 patients (ALT, AST, GGT, ALP, TB) have low specificity, which may be associated with an increase in false positive results. In our study, the diagnosis of liver involvement was established based on data registered at admission, during hospitalization the number of patients with liver involvement may increase in cases with severe evolution and with associated drug-induced liver injury.

## 5. Conclusions

Liver involvement in COVID-19 patients is frequent, usually mild, and occurs mostly in patients over 50 years old, with a higher distribution in the male population. Cholestatic and mixed liver injuries are more frequent than hepatocytolytic injuries. The severity of lung injury evaluated by CT scan, increased values of inflammatory markers (serum ferritin, CRP, fibrinogen), LDH, and low values of SO_2_ can be considered risk factors/predictors for liver injury at admission to hospital.

## Figures and Tables

**Table 1 medicina-58-01714-t001:** Comorbidities in COVID-19 patients with and without liver involvement.

Comorbidity	Group A (n, %)	Group B (n, %)	*p*-Value	Total Study Population (n, %)
Obesity	63 (40.1)	28 (30.4)	0.59	91 (36.5)
Overweight	52 (33.1)	27 (29.3)	0.13	79 (31.7)
Diabetes mellitus type 2	20 (12.7)	13 (14.1)	0.59	33 (13.2)
Arterial systemic hypertension	54 (34.4)	42 (45.6)	0.12	96 (38.5)
Chronic kidney disease	3 (1.9)	3 (3.2)	0.87	6 (2.4)
Chronic obstructive pulmonary disorder	1 (0.6)	2 (2.2)	0.29	3 (1.2)

Group A—patients with liver involvement; Group B—patients without liver involvement.

**Table 2 medicina-58-01714-t002:** Clinical, laboratory and imaging characteristics in COVID-19 patients with and without liver involvement.

Clinical, Laboratory and Imaging Characteristics	Group A (Median, Q1, Q3)	Group B (Median, Q1, Q3)	*p*-Value	OR [CI]
BMI (kg/m^2^)	28.8 [25.7; 32.8]	27.3 [24; 31.6]	0.08	1.04 [0.99; 1.09]
Heart rate (beats/min)	94 [81.5; 104.5]	88 [79.2; 100]	0.03	1.02 [1.002; 1.04]
Systolic blood pressure (mmHg)	130 [118; 141.5]	134 [115.7; 148]	0.62	1.001 [0.99; 1.01]
Diastolic blood pressure (mmHg)	82 [74; 90]	80.5 [72.2; 90]	0.89	1.001 [0.98; 1.02]
Respiratory rate (breaths/minute)	18 [18; 20]	18 [17; 19]	0.005	1.17 [1.05; 1.31]
Saturation (SO_2_)	93 [90; 96]	96 [93; 98]	<0.001	0.86 [0.8; 0.93]
Oxygen flow at admission (L/min)	3 [0; 6]	0 [0; 3]	0.001	1.16 [1.06; 1.26]
Days of hospitalization	9 [7.5; 12]	9 [7; 12]	0.2	1.04 [0.98; 1.1]
CRP (mg/L)	55.2 [21.9; 107]	29.1 [8.5; 84.2]	0.005	1.007 [1.002; 1.01]
Serum ferritin (ng/mL)	708.9 [410.2; 1298.9]	291.1 [141.9; 577.4]	<0.001	1.002 [1.001;1.003]
ESR (mm/h)	42 [20; 60]	36 [20; 53.5]	0.46	1.005 [0.99; 1.02]
IL1 (pg/mL)	1.8 [0.1; 15.3]	2.7 [0; 16.1]	0.58	1.001 [0.99; 1.004]
IL6 (pg/mL)	96.6 [37.9; 183.2]	71.3 [31.8; 150.8]	0.65	1 [1; 1.001]
TNFα (pg/mL)	10.3 [4.5; 17.2]	9.3 [4.4; 21.9]	0.4	1.007 [0.99; 1.02]
WBC (×10^3^/µL)	6.3 [4.8; 8.3]	5.8 [4.8; 7.9]	0.26	1 [1; 1]
Neutrophils (×10^3^/µL)	4.6 [3.1; 6.3]	4.3 [2.9; 6.1]	0.27	1 [1; 1]
Lymphocytes (×10^2^/µL)	9.5 [7; 13]	10.5 [8; 14.2]	0.26	1 [1; 1]
Neu/Ly ratio	5.2 [2.7; 8.3]	4.2 [2.6; 6.3]	0.36	1.03 [0.97; 1.09]
Platelets (×10^3^/µL)	201 [165; 250]	211.5 [163.5; 256.7]	0.38	1 [1; 1]
Hemoglobin (g/dL)	14.2 [13; 15.1]	13.8 [12.9; 14.7]	0.11	1.16 [0.97; 1.38]
D-dimers (ng/mL)	251 [178.5; 329]	209.5 [140; 295.5]	0.52	1 [0.99; 1.001]
PAI-1ng/mL)	301.1 [189.1; 464.8]	300.9 [191.4; 467.8]	0.39	1 [0.99; 1.002]
Fibrinogen (mg/dL)	523 [433.8; 646.2]	470.5 [390.2; 574.7]	0.004	1.003 [1.001; 1.004]
PT (%)	86 [78; 95.3]	84.5 [77.2; 96.1]	0.96	1 [0.98; 1.01]
NtproBNP (ng/L)	21 [6; 50]	27 [10.2; 79.2]	0.35	1 [1; 1]
TnI (ng/mL)	0.3 [0.3; 0.3]	0.3 [0.3; 0.3]	0.71	1.12 [0.6; 2.12]
CK (IU/L)	94 [54; 226]	72 [43; 136]	0.018	1.002 [1; 1.004]
CK-MB (IU/L)	11 [6; 15]	9 [7; 14]	0.07	1.03 [0.99; 1.06]
Myoglobin (ng/mL)	102.1 [72.9; 157]	99.6 [61.5; 160.6]	0.45	0.99 [0.99; 1.001]
Serum proteins(g/dL)	7 [6.5; 7.4]	7.2 [6.6; 7.5]	0.06	0.64 [0.41; 1.02]
Serum albumin (g/dL)	3.9 [3.6; 4.1]	4 [3.8; 4.3]	0.002	0.332 [0.16; 0.67]
LDH (IU/L)	343.5 [271.5; 461.7]	248.5 [213; 319.5]	<0.001	1.007 [1.004;1.009]
ALT (IU/L)	54 [35; 80]	23 [17.7; 31]	<0.001	1.12 [1.08; 1.16]
AST (IU/L)	54.5 [42; 78]	33 [28; 38]	<0.001	1.15 [1.1; 1.19]
GGT (IU/L)	80 [56; 133]	27.5 [23.7; 37]	<0.001	1.16 [1.11; 1.21]
ALP (IU/L)	68 [52.7; 87]	56 [44.5; 67]	<0.001	1.03 [1.02; 1.05]
DB (mg/dL)	0.3 [0.2; 0.4]	0.2 [0.2; 0.4]	0.001	26.3 [3.51; 196.86]
IB (mg/dL)	0.4 [0.2; 0.5]	0.3 [0.2; 0.4]	0.22	2.14 [0.63; 7.26]
TB (mg/dL)	0.7 [0.5; 0.9]	0.6 [0.5; 0.8]	0.008	4.2 [1.47; 12.05]
Serum urea (mg/dL)	33 [26; 41]	33 [27; 43.9]	0.31	1.007 [0.99; 1.02]
Serum creatinine (mg/dL)	0.8 [0.7; 0.9]	0.8 [0.6; 0.9]	0.72	1.11 [0.63; 1.97]
Serum lipase (IU/L)	131.5 [76.2; 208]	142.5 [77.2; 212.7]	0.5	1.001 [0.99; 1.002]
Glycemia (mg/dL)	114.5 [104; 135.5]	108 [99.7; 125.2]	0.2	1.005 [0.99; 1.01]
HbA1c (mg/dL)	5.9 [5.5; 6.4]	5.7 [5.5; 6.2]	0.18	1.28 [0.89; 1.85]
TG (mg/dL)	140 [111.2; 189]	116 [91; 160]	0.012	1.01 [1.002; 1.02]
Serum lipoproteins (mg/dL)	162 [132; 191]	159 [135; 189.5]	0.97	1 [0.99; 1.01]
VLDL (mg/dL)	28 [22; 38.5]	28.5 [20.7; 36]	0.5	0.99 [0.96; 1.02]
LDL (mg/dL)	95 [73; 117.7]	100.5 [79; 120.7]	0.62	0.99 [0.98; 1.01]
HDL (mg/dL)	30 [25.5; 38]	36 [30.7; 40.2]	0.21	0.97 [0.93; 1.02]
Consolidation (% from total lung volume)	1.8 [1.3; 2.8]	1.4 [1.2; 1.7]	<0.001	1.88 [1.34; 2.65]
Mixed lesions (% from total lung volume)	2.3 [1.6; 3.8]	1.7 [1.3; 2.1]	<0.001	1.51 [1.22; 1.85]
Interstitial lesions (% from total lung volume)	43.2 [35.3; 53.3]	34.8 [30.9; 42.5]	<0.001	1.08 [1.05; 1.11]
Normal pulmonary densities (% from total lung volume)	52.5 [40; 60.9]	62 [53.5; 66.2]	<0.001	0.94 [0.91; 0.96]
Total pulmonary lesions (% from total lung volume)	47.5 [39.1; 60]	37.9 [33.7; 46.5]	<0.001	1.07 [1.04; 1.1]
Pulmonary lobes with pneumonia (n)	5 [5; 5]	5 [4; 5]	<0.001	1.69 [1.27; 2.24]
Pulmonary segments with pneumonia (n)	19 [15; 19]	14 [9; 19]	<0.001	1.14 [1.08; 1.2]
RHL (cm)	17.3 [16.2; 18.6]	16.4 [15; 17.5]	<0.001	1.29 [1.12; 1.48]
LHL (cm)	9.1 [7.9; 10.3]	8.4 [7.3; 9.9]	0.02	1.19 [1.02; 1.38]
LD (HU)	45 [29.1; 51.6]	49.8 [42.8; 55.3]	<0.001	0.96 [0.94; 0.98]

ALP, alkaline phosphatase; ALT, alanine transaminase; AST, aspartate transaminase; BMI, body mass index; CK, creatine kinase; CK-MB, creatine kinase MB isoform; CRP, C-reactive protein; DB, direct bilirubin; ESR, erythrocyte sedimentation rate; GGT, gamma-glutamyl transaminase; HbA1c, glycosylated hemoglobin; HDL, high-density lipoprotein; IB, indirect bilirubin; IL-1, interleukin-1; IL-6, interleukin-6; LD, liver density; LDH, lactate dehydrogenase; LDL, low-density lipoprotein; LHL, antero-posterior diameter of left hepatic lobe; Neu/Ly, neutrophil-to-lymphocyte ratio; NT-proBNP, N-terminal pro-B-type natriuretic peptide; PAI-1, plasminogen activator inhibitor-1; PT, prothrombin time; RHL, cranio-caudal diameter of right hepatic lobe; TB, total bilirubin; TNFα, tumor necrosis factor alpha; TG, serum tryglicerides; TnI, troponin I; VLDL, very-low-density lipoprotein; WBC, white blood cells.

**Table 3 medicina-58-01714-t003:** Correlations of clinical, laboratory and imaging parameters with liver involvement.

Clinical, Laboratory and Imaging Characteristics	Pearson Correlation	*p*-Value
BMI	0.115	0.08
Heart rate	0.14	0.03
Systolic blood pressure	0.034	0.59
Diastolic blood pressure	0.009	0.89
Respiratory rate	0.187	0.001
Oxygen saturation	−0.27	<0.001
Oxygen flow at admission	0.227	0.001
Days of hospitalization	0.084	0.19
CRP	0.19	0.004
Serum ferritin	0.36	<0.001
ESR	0.06	0.47
IL1	0.05	0.56
IL6	0.03	0.65
TNFα	0.06	0.39
WBC	0.07	0.26
Neutrophils	0.07	0.27
Lymphocytes	−0.07	0.25
Neu/Ly ratio	0.06	0.36
Platelets	0.06	0.37
Hemoglobin	0.1	0.11
D-dimers	0.06	0.38
PAI-1	0.04	0.38
Fibrinogen	0.187	0.003
PT	−0.003	0.97
NtproBNP	−0.07	0.29
TnI	0.02	0.71
CK	0.16	0.01
CK-MB	0.12	0.06
Myoglobin	−0.05	0.45
Serum proteins	−0.135	0.06
Serum albumin	−0.216	0.002
LDH	0.332	<0.001
ALT	0.327	<0.001
AST	0.383	<0.001
GGT	0.421	<0.001
ALP	0.287	<0.001
DB	0.22	0.001
IB	0.08	0.22
TB	0.184	0.006
Serum urea	0.07	0.29
Serum creatinine	0.02	0.72
Serum lipase	0.05	0.49
Glycemia	0.08	0.19
HbA1c	0.11	0.18
TG	0.219	0.009
Serum lipoproteins	−0.003	0.97
VLDL	−0.1	0.49
LDL	−0.06	0.62
HDL	−0.14	0.21
Consolidation (% from total lung volume)	0.247	<0.001
Mixed lesions (% from total lung volume)	0.268	<0.001
Interstitial lesions (% from total lung volume)	0.335	<0.001
Normal pulmonary densities (% from total lung volume)	−0.337	<0.001
Total pulmonary lesions (% from total lung volume)	0.337	<0.001
Pulmonary lobes with pneumonia	0.249	<0.001
Pulmonary segments with pneumonia	0.311	<0.001
RHL	0.23	<0.001
LHL	0.144	0.023
LD	−0.269	<0.001

**Table 4 medicina-58-01714-t004:** Symptomatology of SARS-CoV-2 patients with and without liver involvement.

Symptoms	Group A (n, %)	Group B (n, %)	*p*-Value
fever	126, (80.3)	58, (63)	0.003
chills	70, (44.6)	47, (51.1)	0.32
cough	128, (81.5)	82, (89.1)	0.1
nasal congestion	11, (7)	8, (8.7)	0.63
rhynorrhea	7, (4.5)	8, (8.7)	0.18
dysphagia	9, (5.7)	10, (10.9)	0.15
anosmia	16, (10.2)	14, (15.2)	0.24
ageusia	16, (10.2)	11, (12)	0.63
asthenia	90, (57.3)	59, (64.1)	0.29
myalgia	71, (45.2)	52, (56.5)	0.08
arthralgia	38, (24.2)	22, (23.9)	0.96
thoracic pain	32, (20.4)	15, (16.3)	0.41
abdominal pain	11, (7)	3, (3.3)	0.2
diarrhea	44, (28)	14, (15.2)	0.02
loss of appetite	44, (28)	32, (34.8)	0.28
nausea	31, (19.7)	17, (18.5)	0.8
vomiting	16, (10.2)	5, (5.4)	0.18
headache	71, (45.2)	28, (30.4)	0.02
paraesthesia	1, (0.6)	1, (1.1)	0.7
confusion	3, (1.9)	2, (2.2)	0.89
conjunctival hyperemia	2, (1.3)	1, (1.1)	0.89
dyspnea	67, (42.7)	34, (37)	0.37
polypnea	33, (21)	7, (7.6)	0.004
respiratory insufficiency	106, (67.5)	39, (42.4)	<0.001

**Table 5 medicina-58-01714-t005:** Characteristics of liver involvement in patients with COVID-19.

Variables		On Admission (n, % Study Population)
ALT	1–3 × ULN	79 (31.7)
	3–5 × ULN	8 (3.2)
	5–10 × ULN	0 (0)
	≥10 × ULN	1 (0.4)
AST	1–3 × ULN	67 (26.9)
	3–5 × ULN	2 (0.8)
	5–10 × ULN	2 (0.8)
	≥10 × ULN	0 (0)
GGT	1–2 × ULN	69 (27.7)
	2–4 × ULN	42 (16.9)
	4–8 × ULN	12 (4.8)
	≥8 × ULN	6 (2.4)
ALP	1–2 × ULN	10 (4)
	2–4 × ULN	0 (0)
	4–8 × ULN	0 (0)
	≥8 × ULN	0 (0)
TB	1–2 × ULN	13 (5.2)
	2–4 × ULN	0 (0)
	4–8 × ULN	0 (0)
	≥8 × ULN	0 (0)
Liver involvement	Grade 1	94 (37.7)
	Grade 2	43 (17.2)
	Grade 3	14 (5.6)
	Grade 4	6 (2.4)
Hepatocytolytic involvement		26 (10.4)
Cholestatic involvement		63 (25.3)
Mixed liver involvement		68 (27.3)

**Table 6 medicina-58-01714-t006:** Characteristics of abnormal liver function/injuryin COVID-19 patients (grade 1—patients with abnormal liver function and grade 4—patients with liver injury).

Comorbidities, Clinical, Laboratory and Imaging Characteristics	Grade 1 (Abnormal Liver Function) (Median, Q1, Q3)/(n%)	Grade 4 (Liver Injury) (Median, Q1, Q3) (n,%)
Age (years)	54 [44; 65]	57 [43.5; 66]
Male	59 (63.4)	6 (100%)
Female	34 (36.6)	0 (0%)
Obesity (n)	42 (45.1)	1 (16.6)
Overweight (n)	25 (26.9)	5 (83.4)
Diabetes mellitus type 2 (n)	9 (9.7)	2 (33.3)
Arterial systemic hypertension (n)	37 (39.8)	3 (50)
Chronic kidney disease (n)	2 (2.1)	0 (0)
Chronic obstructive pulmonary disorder (n)	0 (0)	0 (0)
BMI (kg/m^2^)	29.4 [25.7; 34.2]	27.4 [25.7; 30.2]
Heart rate (beats/min)	94 [81; 103]	87 [78; 103.7]
Systolic blood pressure (mmHg)	128 [117; 141]	138 [110.5; 158.5]
Diastolic blood pressure (mmHg)	80 [71; 90]	88.5 [73; 97.7]
Respiratory rate (breaths/minute)	18 [17; 20]	18 [17; 19.2]
Saturation (SO_2_)	93 [90; 96]	92 [91.2; 95.2]
Oxygen flow at admission(L/min)	3 [0; 4.5]	4 [0; 7.5]
Days of hospitalization	8 [6.7; 9.2]	10 [8; 13]
CRP (mg/L)	45.4 [18.5; 106]	67 [35.6; 175.5]
Serum ferritin (ng/mL)	610.7 [351.1; 1321.6]	1072 [662.3; 1460]
ESR (mm/h)	42 [18.5; 60]	52 [44; 52]
IL1 (pg/mL)	1.7 [0.2; 12.1]	1.7 [0.8; 15.3]
IL6 (pg/mL)	88.6 [39.4; 150.4]	104.2 [66; 189.5]
TNFα (pg/mL)	11.7 [5.7; 20.4]	11 [4.3; 29.2]
WBC (×10^3^/µL)	6.2 [4.8; 8.4]	7.6 [5.6; 11]
Neutrophils (×10^3^/µL)	4.4 [2.9; 6.7]	6 [4.1; 9.1]
Lymphocytes (×10^2^/µL)	10 [7; 13]	11 [8; 16.5]
Neu/Ly ratio	4.7 [2.7; 8.3]	6.1 [3.1; 8.7]
Platelets (×10^3^/µL)	197 [162.5; 234.5]	290 [198; 416.2]
Hemoglobin (g/dL)	14 [12.9; 15]	14.1 [13.6; 14.9]
D-dimers (ng/mL)	257 [175; 395.5]	236.5 [127.7; 378.5]
PAI-1(ng/mL)	314.5 [192.5; 478.3]	603.6 [361.7; 754.7]
Fibrinogen (mg/dL)	508 [425; 600.5]	685 [552.2; 815.7]
PT (%)	87.5 [78.3; 95.5]	83.5 [78.8; 87.8]
NtproBNP (ng/L)	20 [6; 44.7]	39 [5.7; 1259.7]
TnI (ng/mL)	0.03 [0.03; 0.03]	0.03 [0.03; 0.08]
CK (IU/L)	89 [55; 226]	45.5 [33.5; 84.2]
CK-MB (IU/L)	11 [6; 15]	8 [4.5; 14]
Myoglobin (ng/mL)	95.3 [70; 157.8]	116.5 [66.2; 145.2]
Serum proteins(g/dL)	7 [6.5; 7.4]	7.2 [6.9; 7.8]
Serum albumin (g/dL)	3.9 [3.5; 4.1]	3.9 [3.6; 3.9]
LDH (IU/L)	330 [262.2; 441.5]	460 [302.2; 606.5]
ALT (IU/L)	51 [32; 63]	94.5 [39.2; 337.5]
AST (IU/L)	49 [40; 70]	90.5 [66; 183.5]
GGT (IU/L)	60 [50; 76]	500 [399.7; 612.2]
ALP (IU/L)	62 [50; 76]	118 [104; 134.5]
DB (mg/dL)	0.3 [0.2; 0.4]	0.5 [0.4; 0.6]
IB (mg/dL)	0.4 [0.2; 0.5]	0.4 [0.1; 0.6]
TB (mg/dL)	0.6 [0.5; 0.9]	0.9 [0.7; 0.9]
Serum urea (mg/dL)	34.5 [28.2; 42.5]	36 [28; 74.2]
Serum creatinine (mg/dL)	0.8 [0.7; 0.9]	0.8 [0.6; 1.3]
Serum lipase (IU/L)	113 [73; 187]	169 [127.5; 216]
Glycemia (mg/dL)	115.5 [104; 134.7]	146 [109; 196.5]
HbA1c (mg/dL)	5.8 [5.5; 6.3]	6.1 [5.7; 9.1]
TG (mg/dL)	137 [115; 182]	177 [103; 177]
Serum lipoproteins (mg/dL)	158 [135; 186]	162.5 [134; 162.5]
Consolidation (% from total lung volume)	1.8 [1.3; 2.6]	1.8 [1.4; 2.9]
Mixed lesions (% from total lung volume)	2.3 [1.6; 3.6]	2.5 [1.9; 4.9]
Interstitial lesions (% from total lung volume)	42.2 [35.2; 51.8]	43.1 [33.6; 54.1]
Normal pulmonary densities (% from total lung volume)	54.1 [40.1; 60.9]	52.5 [37.9; 63]
Total pulmonary lesions (% from total lung volume)	45.9 [39; 59.9]	47.4 [36.9; 62]
Pulmonary lobes with pneumonia (n)	5 [5; 5]	5 [5; 5]
Pulmonary segments with pneumonia (n)	17 [15; 19]	19 [17.2; 19]
RHL (cm)	17.3 [16.1; 18.6]	17.6 [16.3; 18.1]
LHL (cm)	9.1 [7.5; 10.3]	17.6 [16.3; 18.1]
LD (HU)	42.6 [29.3; 51.9]	42 [31.7; 48.3]

ALP, alkaline phosphatase; ALT, alanine transaminase; AST, aspartate transaminase; BMI, body mass index; CK, creatine kinase; CK-MB, creatine kinase MB isoform; CRP, C-reactive protein; DB, direct bilirubin; ESR, erythrocyte sedimentation rate; GGT, gamma-glutamyl transaminase; HbA1c, glycosylated hemoglobin; IB, indirect bilirubin; IL-1, interleukin-1; IL-6, interleukin-6; LD, liver density; LDH, lactate dehydrogenase; LHL, antero-posterior diameter of left hepatic lobe; Neu/Ly, neutrophil-to-lymphocyte ratio; NT-proBNP, N-terminal pro-B-type natriuretic peptide; PAI-1, plasminogen activator inhibitor-1; PT, prothrombin time; RHL, cranio-caudal diameter of right hepatic lobe; TB, total bilirubin; TNFα, tumor necrosis factor alpha; TG, serum tryglicerides; TnI, troponin I; WBC, white blood cells.

## Data Availability

The data presented in this study are available on request from the corresponding authors.

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
