# Peer review of "Predictors of Liver Injury in Hospitalized Patients with SARS-CoV-2 Infection"

_medicina, 2022, doi:10.3390/medicina58121714_

Round 1

Reviewer 1 Report

Thanks for the opportunity to review this very interesting work.  Even though the topic is pertinent and interesting, there are some significant methodological and interpretation bias that merit attention.

Specifically:

Results:

Pg6, line 175-190: based on the tabulated results (table 2), there is no double risk on any of the risk factors, so I would review statistical results and revise accordingly.

Pg7, table 3:  LFTs cannot be both indicators of liver involvement grading and AND risk factors. It is a tautology. I would therefore remove them from the correlation analysis. [Similar argument may be made for the acute phase proteins  (CRP and ferritin) and fibrinogen, since they are all produced by the liver and therefore might be affected during a hepatic insult. However, you have not included these variables as part of the criteria for diagnosing liver involvement, so it would be acceptable to keep in the analysis on the " risk factor" side of the analysis.]

Pg8, line 211-212: Was there significant difference in the overall mortality between the two groups? 

Discussion:

Pg10, line 249-253. ALT actually has high specificity for the liver since it is only found in the liver (and the kidney).

Pg11, line 276-284. Even though clotting factors seem to promote steatosis in experiments on mice [25], it is unclear if this is the case in humans. 

Pg 11, line 279-283. The authors state that "the authors" (i.e. Chen et al) "concluded that hepatic steatosis is probably a self-limiting condition related to COVID-19". I am not certain whether Chen et al conveyed such message in their study [26]. Such statement lacks causation: it is unclear if these pts had been steatotic prior to their COVID infection rather than having developed steatosis while infected by COVID. I am not aware of such evidence ( i.e. that acute COVID infection can cause de novo steatosis); even though it is possible to be positively correlated with COVID disease severity (as a surrogate indicator of adiposity, a well known risk factor for COVID-related mortality and cytokine storm release in general); and vice versa, there is no causative evidence that steatosis identified during the course of COVID (and probably pre-existing) improved due to resolution of COVID infection rather than inadequate caloric intake during the course of the critical illness (and resultant increased hepatic lipolysis, a very well established response in acute metabolic state, known to be taking place during fasting/ decreased caloric intake/ increased metabolic demands status). I would therefore revise this paragraph or delete it entirely.

Pg11, line 294. " aggravate coagulopathy". The authors probably meant hypercoagulability or intrahepatic clot development risk.

Pg 11, line 310-314. This statement is not supported by the produced Pearson analysis. 

Thanks again for the opportunity to review your work. 

Author Response

Thank you for giving us the opportunity to submit a revised draft of the manuscript “Pericardial involvement in COVID-19 patients” for publication in “Medicina”. We appreciate the time and effort that you dedicated to providing feedback on our manuscript. The authors have carefully considered the comments, and tried our best to address every one of them; we have incorporated all the suggestions in the manuscript. Please find the attached file with our point-by-point response to your comments and concerns.

Reviewer 2 Report

The authors in this manuscript submission present a retrospective cohort study evaluating the impact of COVID-19 infection on the liver and the risk factors/predictors for liver injury on 249 patients at admission to the hospital. Study cohorts were divided into Group A (157 patients with liver involvement) and Group B (92 patients without liver involvement), and demographic and lifestyle parameters, anthropometric parameters, comorbidities, clinical parameters, inflammation markers, complete blood count, coagulation, and biochemical parameters were recorded. Lung parenchyma, liver dimensions, and morphology were evaluated by CT scans. Authors report that patients with liver involvement had higher heart and respiratory rates, lower SO2, and necessitated higher oxygen flow at admittance. Higher serum levels of C-reactive protein, fibrinogen, ferritin, creatine kinase, lactate dehydrogenase (LDH), serum triglycerides, and lower values for serum albumin were found in the patients with liver involvement. These patients presented more extensive lung injury with higher percentages of alveolar, mixed, and interstitial lesions, an increase in liver dimensions, and lower density ranges for the liver parenchyma. Authors conclude that liver involvement in COVID-19 patients is frequent, usually mild, and occurs mostly in male patients over 50 years old. Cholestatic and mixed liver injuries are more frequent than hepatocytolytic injuries. The severity of lung injury evaluated by CT scan, increased values of inflammatory markers, LDH, and low values of SO2 can be considered risk factors/predictors for liver injury at admission to the hospital.

The current study does expand upon the previous studies that demonstrated the association of COVID-19 to liver injury, specifically to increased liver enzymes such as ALT, AST, alkaline phosphatase (ALP), and glutamyl transferase (GGT) (Phipps et al, Hepatol 2020; Bertolini et al, Hepatol 2020, Goyal et al, N Engl J Med 2020; Cai et al, J Hepatol 2020 etc.). Serological inflammatory markers such as ferritin and IL-6 have been significantly associated with severe liver injury controlling for age, sex, and peak levels of D-dimer, CRP, CK and troponin (Phipps et al, Hepatol 2020). Although it is thought to be a combination of direct virally mediated injury as well as the immune-mediated inflammatory response, the mechanism by which SARS-CoV2 impacts the liver is not clearly understood. Moreover, the risk factors and predictors of liver injury in hospitalized patients are not clearly understood. The current study implicates the severity of lung injury evaluated by CT scan, LDH (associated with severe COVID-19; Henry et al, Am J Emerg Med, 2020; Han et al, Aging (Albany NY), 2020), and low values of SO2, in addition to increased values of inflammatory markers, as the risk factors/predictors for liver injury at admission to the hospital. In light of these publications and numerous others, the novelty of the study is not clear, however, this is a straightforward descriptive retrospective study that suggests risk factors/predictors of liver injury (e.g., the correlation of lung injury evaluated by CT scan with the hepatic involvement, superior to liver measurements (RHL, LHL dimensions, and liver density) in hospitalized patients with SARS-CoV2 infection. 

Major Concerns:

1. The authors have divided 249 patients into 2 groups (Group A, 157 patients with liver involvement; and Group B, 92 patients without liver involvement). Considering the levels of liver injury enzymes (e.g., ALT, AST, ALP, GGT, and TB), the liver involvement was further graded into 4 grades (grade 1 to 4). Grade 1 corresponds to abnormal liver functions and grade 2-4 correspond to liver injury. However, the description of patients without liver involvement is not clear. What defines “without liver involvement”? It is better to define liver involvement (Group A) as abnormal liver function/injury; and without liver involvement (Group B) as without abnormal liver function/injury. It needs clarification if hepatocytolytic or cholestatic involvement were included in Group B? 

2. The study includes exactly 157 males and 92 females, and therefore should not be confused males as Group A and females as Group B. It needs clarity in “Characteristics of the study participants”.      

3. It will be interesting to compare grade 1 to 4 liver involvement patients with lung injury, SO2, CRP, ferritin, fibrinogen, LDH, inflammatory cytokines, and other variables such as age, sex, obesity, and diabetes etc. This might help associate the abnormal liver function vs liver injury (or severe liver injury) to other clinical outcomes.   

4. The introduction does not provide sufficient background and relevant references. It should be revised to include these information and highlight the rationale of the study.

Minor concerns:

1. In Materials and Methods, the institutional approval of the study by the local Ethics Committee of the Institute and the informed consent should be included after the first mention of 249 patients in the methods section.

Author Response

Thank you for giving us the opportunity to submit a revised draft of the manuscript “Predictors of liver injury in hospitalized patients with SARS-CoV-2 infection” for publication in “Medicina”. We appreciate the time and effort that you dedicated to providing feedback on our manuscript. The authors have carefully considered the comments, and tried our best to address every one of them; we have incorporated all the suggestions in the manuscript. Please find the attached file with our point-by-point response to your comments and concerns.

Round 2

Reviewer 2 Report

I appreciate authors addressing all the comments and revised manuscripts accordingly.